# Structure and mechanism of the Nap adhesion complex from the human pathogen *Mycoplasma genitalium*

David Aparicio[1,4], Margot P. Scheffer[2,4], Marina Marcos-Silva[3,4], David Vizarraga[1], Lasse Sprankel[2], Mercè Ratera [1], Miriam S. Weber [2], Anja Seybert[2], Sergi Torres-Puig[3], Luis Gonzalez-Gonzalez [3], Julian Reitz[2], Enrique Querol[3], Jaume Piñol[3], Oscar Q. Pich [3✉], Ignacio Fita[1✉] & Achilleas S. Frangakis [2✉]

*Mycoplasma genitalium* is a human pathogen adhering to host target epithelial cells and causing urethritis, cervicitis and pelvic inflammatory disease. Essential for infectivity is a transmembrane adhesion complex called Nap comprising proteins P110 and P140. Here we report the crystal structure of P140 both alone and in complex with the N-terminal domain of P110. By cryo-electron microscopy (cryo-EM) and tomography (cryo-ET) we find closed and open Nap conformations, determined at 9.8 and 15 Å, respectively. Both crystal structures and the cryo-EM structure are found in a closed conformation, where the sialic acid binding site in P110 is occluded. By contrast, the cryo-ET structure shows an open conformation, where the binding site is accessible. Structural information, in combination with functional studies, suggests a mechanism for attachment and release of *M. genitalium* to and from the host cell receptor, in which Nap conformations alternate to sustain motility and guarantee infectivity.

[1] Instituto de Biología Molecular de Barcelona (IBMB-CSIC) and Maria de Maeztu Unit of Excellence, Parc Científic de Barcelona, Baldiri Reixac 10, 08028 Barcelona, Spain. [2] Buchmann Institute for Molecular Life Sciences, Max-von-Laue Str. 15, 60438 Frankfurt, Germany. [3] Institut de Biotecnologia i Biomedicina and Departament de Bioquímica i Biologia Molecular, Universitat Autònoma de Barcelona, Bellaterra, 08193 Barcelona, Spain. [4] These authors contributed equally: David Aparicio, Margot P. Scheffer, Marina Marcos-Silva. ✉email: oquijada@tauli.cat; ifrcri@ibmb.csic.es; achilleas.frangakis@biophysik.org

The human pathogen *Mycoplasma genitalium*, a member of the *pneumoniae* cluster of mycoplasmas, binds to eukaryotic cells by means of its adhesion complex, the Nap. This complex is formed by two heterodimers, each consisting of proteins P110 and P140[1–6]. In addition to their roles in cytadherence and motility, P110 and P140 are immunodominant proteins and constitute the main target of host antibodies during infection[7–9]. Antibiotic resistance to human pathogens from the *pneumoniae* cluster[10–13] is increasing at an alarming rate, making it necessary to explore novel therapeutic strategies. Anti-adherence molecules, aimed at preventing the establishment of infection, are attractive potential antimicrobial drugs[14,15]. A deep understanding of the Nap structure and adhesion mechanism will facilitate the development of anti-adherence therapies. Recently, we determined the crystal structure of the extracellular region of P110 and demonstrated its binding to sialic acid receptors[6]. Here, we address the structure and mechanism of the Nap adhesion complex and reveal an intricate interplay between P110 and P140.

## Results

**Crystal structure of P140 and in complex with P110N.** Crystals were obtained from the extracellular region of P140 (residues 23–1351) (Fig. 1, Supplementary Figs. 1 and 2), both alone and in complex with the N-terminal domain of P110 (P110N: residues 23–827) (Fig. 2a, Supplementary Fig. 2). The structure of P140, for which there are no molecular models or experimental phases available, was determined by density modification techniques, starting with a mask derived from the sub-tomogram-averaged map of the whole Nap obtained by cryo-electron tomography (cryo-ET) (see Methods). With four heterodimers in the asymmetric unit, the P140–P110N crystals were refined at 2.65 Å resolution to a final model with agreement $R$ and $R_{free}$ factors of 18.7 and 22.4, respectively (Supplementary Table 1).

**Structural similarities of P140 and P110.** The structure of the extracellular region of P140, with a bulky N-terminal domain (residues 23–1243) and a small C-terminal domain (residues 1244–1351), has an overall shape resembling the capital letter P (Fig. 1a). The N-terminal domain consists of a seven-bladed (β-sheet) propeller and a "crown" formed by the clustering of the long polypeptide segments that emerge from the propeller (Supplementary Fig. 3). β-Sheets I to VI each have four strands, while β-sheet VII, the last β-sheet in the propeller that connects directly with the C-terminal domain, has only two strands (Fig. 1b). P140 and P110 share many features of domain organization and the topology of the secondary structural elements, suggesting a common ancestor, although the degree of conservation differs for the N-terminal and C-terminal domains (Supplementary Fig. 4). The N-terminal domains, with an RMSD of 3.5 Å between the Cα atoms of 359 structurally equivalent residues (~28%), are markedly different in the crown. In contrast, the C-terminal domains are closely related, with 74 equivalent residues (~71%) and an RMSD of 2.2 Å.

**Binding site analysis of the Nap.** The binding site for the sialylated oligosaccharides, identified in the structure of P110[6], is located at the interface between P140 and P110 in the crystal structure of the P140–P110N complex (Fig. 2a–d). Interaction of the two subunits changes the position of the sialic binding β-hairpin while the interfering loop (P140 residues 807–827) inserts into the binding pocket of P110N, sterically interfering with the binding of oligosaccharides to the complex (Fig. 2c–d). In agreement with this, surface plasmon resonance analysis shows that, in solution, sialylated compounds 3SL and 6SL (neuraminic acid forming an α2–3 or an α2–6 linkage to a lactose monosaccharide, respectively), which bind to P110 alone[6], do not bind either to P140 alone or to P140–P110 complexes (Supplementary Fig. 5). The structure of the P140–P110N complex suggests that P110 residues Gln460–Asp461, from the binding β-hairpin, and

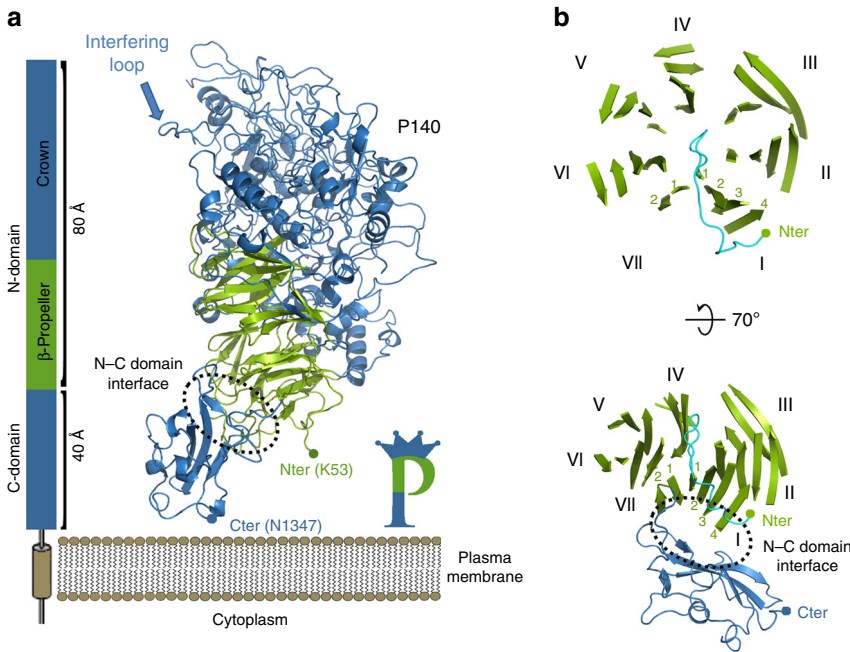

**Fig. 1 Structure of P140. a** P140 contains a large N-terminal domain and a smaller C-terminal domain, which is followed in sequence by a predicted transmembrane helix. The N-terminal domain has two distinct regions: the β-propeller (green) and the crown (blue). **b** Two views of the β-propeller, 70° apart. The first and last β-sheets of the propeller (I and VII, respectively) interact with the C-terminal domain and are structurally contiguous in the propeller ring. A large β-bulge (residues 69–77, in cyan) occludes the center of the propeller ring. The presence of only two strands in the last β-sheet (VII) constrains the interface between the N-terminal and C-terminal domains.

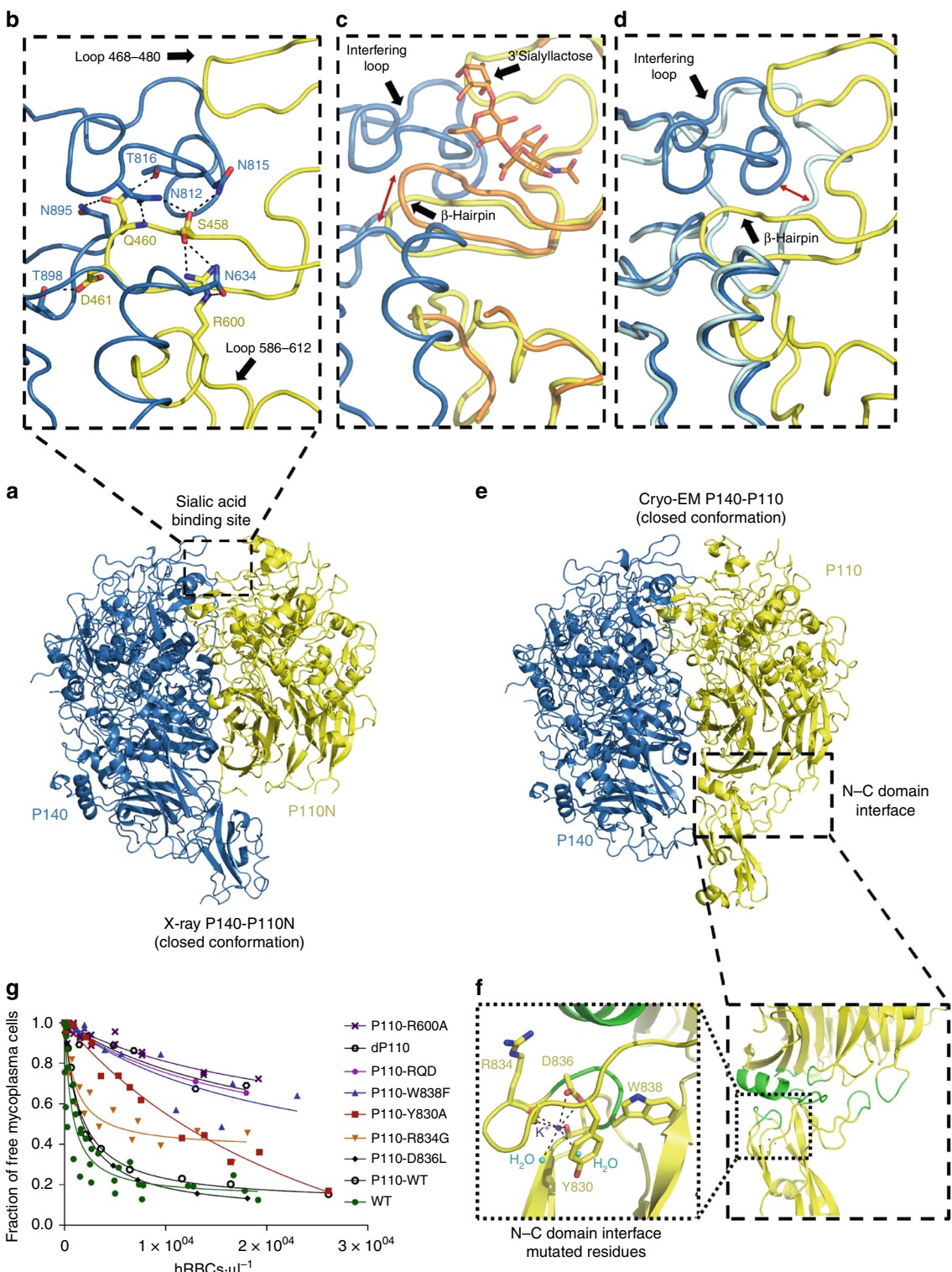

Arg600, close to the receptor-binding site, play an important role in the interaction with P140 (Fig. 2b, d). P110 variants carrying an R600A substitution or the triple substitution RQD (R600A, Q460A, and D461A) were introduced by transposon delivery into an *M. genitalium* P110 null mutant[1]. Strains expressing the P110-RQD variant protein, which was barely detectable, showed a null binding capacity phenotype (Fig. 2g, Supplementary Fig. 6b). The variant protein P110-R600A was well expressed, but the strain presented no capacity for adherence and characterization of cell motility was not feasible.

**Single-particle cryo-EM of the P140–P110 extracellular region.** Using a sample of P140–P110 complexes, with the complete extracellular region included for both subunits (P140 residues 23–1351 and P110 residues 23–938), we performed single-particle cryo-electron microscopy (cryo-EM). We obtained a map with an overall resolution of 4.1 Å, although non-isotropic (Fig. 2e, Supplementary Table 2, Supplementary Figs. 7 and 8). The P140–P110N X-ray structure could be fitted as a rigid-body without modifications into the P140–P110 cryo-EM map with UCSF Chimera[16] (Supplementary Table 3, Supplementary Fig. 7,

**Fig. 2 Closed conformation of the P140–P110 heterodimers by X-ray and cryo-EM. a** X-ray crystal structure of the P140–P110N heterodimer. **b** Network of hydrogen bonds in the vicinity of the oligosaccharide-binding site between P140 (blue) and P110N (yellow). Rearrangements experienced by P110N (**c**) and P140 (**d**) with respect to the structures of P110 alone (orange) and P140 alone (light blue), respectively. Rearrangements of the ß-hairpin and the interfering loop are indicated with red arrows. Oligosaccharide 3SL (**c**) is depicted as it was found in the structure of P110 alone[6], showing that it would have steric clashes in the closed conformation of the P140–P110N complex. **e** Cryo-EM structure of the P140–P110 heterodimer. The C-terminal domain of P140 is missing in the structure because it was not defined in the cryo-EM map (Supplementary Fig. 7). The C-terminal domain of P110 was visible in the cryo-EM map, although with a local resolution lower than that for the N-terminal domains. **f** Details of the interface (depicted in green) between the N-terminal and C-terminal domains, close to where a potassium ion was found in the structures of P110 alone. **g** Hemadsorption assays by fluorescence-activated cell sorting (FACS) analysis. Graphic represents inverse Langmuir plots (see also Supplementary Information) containing a fixed amount of mycoplasma cells and increasing amounts of human red blood cells (hRBCs). Plots represent the best-fitting curves to a series of hemadsorption measurements obtained from at least two biological repeats for each strain. We performed a double-gating strategy, using a preliminary FL3-H/FL2-H gate following an SSC-H/FL1-H gate. Mutations are either in the vicinity of the oligosaccharide-binding site at the interface between P140 and P110 (residues explicitly shown in panel **b**) or close to the interface between the N-terminal and C-terminal domains of P110 (residues shown in panel **f**). $R^2$ for each fitting curve: WT (0.8576), dP110 (0.9054), P110-WT (0.9921), P110-Y830A (0.9663), P110-W838F (0.8339), P110-R834G (0.8757), P110-D836L (0.9993), P110-R600A (0.8663), and P110-RQD (0.9999).

Supplementary Movie 1). Therefore, the structure of the P140–P110 complex found by cryo-EM corresponds to the conformation of the X-ray P140–P110N structure, where access to the sialylated oligosaccharides binding site is occluded (Supplementary Movie 1). In the P140–P110 cryo-EM map, there is no density for the C-terminal domain of P140, whereas the C-terminal domain of P110, which is absent in the P140–P110N complex, is visible, albeit with a weak density (Fig. 2e, Supplementary Fig. 7). This indicates a significant flexibility of the C-terminal domains with respect to the bulkier N-terminal domains. In the P140–P110 complex found by cryo-EM, the interface between subunits spans 2758 Å² with 30 hydrogen bonds and an estimated Gibbs free energy of −20 kcal/mol, resulting in a 100% probability of the formation of the complex (PISA server[17]).

**Motility analysis of mutants**. Five mutations were introduced close to the interface between the N-terminal and C-terminal domains, to check if either adhesion or motility was affected. The five mutated residues (Y830A, R834G, D836L, W838F, and G839F) were chosen in the vicinity of the potassium-binding site found in P110[6] (Fig. 2e, f, Supplementary Table 4). The P110-G839F strain showed no detectable levels of adhesins, while P110-D836L exhibited hemadsorption values similar to those of wild-type cells (Fig. 2g). The P110-Y830A and P110-R834G variants showed intermediate hemadsorption phenotypes, characterized by unusually high $B_{max}$ values (see Methods). Hemadsorption of the P110-W838F mutant was severely impaired, with high $K_d$ and $B_{max}$ values. Motility was examined by time-lapse microcinematography to monitor the movements of individual cells for 120 s (Supplementary Fig. 6, Supplementary Table 5). In the P110-W838F mutant, cells were completely non-motile and, in agreement with this extreme phenotype, phase contrast images revealed the presence of large aggregates resulting from the inability of cells to spread. The Y830A, R834G, and D836L variants also exhibited altered gliding velocities, indicating that structural integrity at the interface between the N-terminal and C-terminal domains, away from the cell receptor-binding site, is critical for motility in *M. genitalium*.

**Single-particle analysis of the Nap**. Next, we performed single-particle cryo-EM using a purified sample of Nap complexes, obtained as previously described[3] (see Methods). The purified Nap complexes contain full-length P140 and P110 proteins, including the transmembrane helices and cytoplasmic regions, which are required for formation of tetramers. Classification indicated that only ~15–20% of the images corresponded to

complete Nap particles where the extracellular region was well defined. The best map, with an overall resolution of 9.8 Å, allowed accurate rigid-body fitting with Chimera[16] of the P140 structure alone and the P110 structure alone, confirming the presence of a dimer of P140–P110 heterodimers in each Nap (Fig. 3a–c, Supplementary Fig. 9, Supplementary Table 2). The arrangement of P140–P110 heterodimers is essentially identical to the X-ray and cryo-EM P140–P110 structures, with the interactions between both subunits preventing access to the sialylated oligosaccharide-binding site. This is the "closed" conformation of the Nap (Supplementary Movie 2). C-terminal domains act like stalks connecting the large N-terminal domain of each subunit with the outer surface of the cell membrane. The two P110 subunits are almost parallel to the dimer axis and face away from each other, while the two P140 subunits of the Nap adopt a "V-shaped" arrangement with the C-terminal domains very close to each other (Fig. 3c). The distance between heterodimers is large (Fig. 3b, c), suggesting that their interaction is weak. This is in agreement with MALS measurements where only heterodimers are detected when mixing equimolecular amounts of constructs from the extracellular regions of P110 and P140 (Supplementary Fig. 10). Therefore, there are two interfaces in the Nap between the extracellular regions of P140 and P110, which we name the "tight" and "loose" interfaces (Fig. 3b, e). The cryo-EM map of the Nap shows density corresponding to the Nap transmembrane region and the positions of the transmembrane helices (their N-terminal ends) can be seen (Fig. 3a, c). The close proximity of the P140 C-terminal domains also brings the N-terminal ends of the corresponding transmembrane helices closer together. For P140 and P110, sequence analysis indicates the presence of one transmembrane helix containing one and two Engelman motifs (GXXXG, with X any residue, in general hydrophobic), respectively, which are characteristic of high-affinity interactions in membrane helices[18,19].

**Cryo-electron tomography of the in situ Nap**. The structure of the in situ Nap complex was also determined by cryo-ET from mildly lysed *M. genitalium* cells[3]. Classification of Nap volumes and subsequent sub-tomogram averaging provided an improved map for the most abundant class (~85%) at 15 Å resolution, in which the four subunits, two from P110 and two from P140, are clearly distinguished and present a nearly perfect twofold symmetry (Fig. 3d–f, Supplementary Fig. 11). The rigid-body fitting of the structures of P110 alone and P140 alone into the cryo-ET density with Chimera[16] reveals major differences from the cryo-EM structure of the Nap (Fig. 3, Fig. 4a, Supplementary Table 3). In the cryo-ET structure, the longest axis of the four subunits runs parallel to the axis of twofold symmetry of the Nap. This

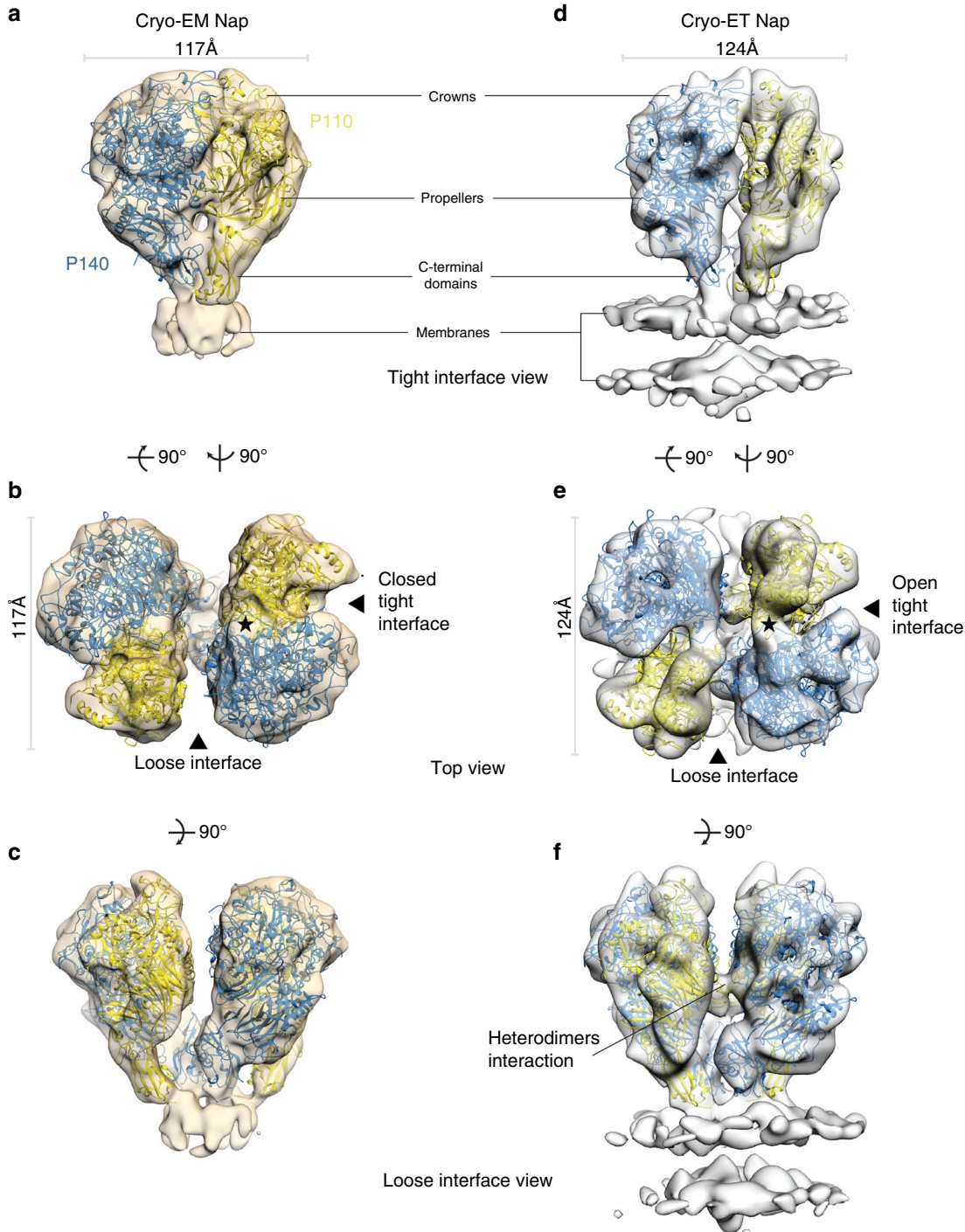

**Fig. 3 The structure of Nap by single-particle cryo-EM and in situ cryo-ET. a–c** Three different views of the single-particle cryo-EM from the Nap (beige surface) with the structures of P140 (blue) and P110 (yellow) fitted into the density. In **a** the crown, propeller, and C-terminal domain are indicated. **b** Top view of the Nap, depicting the loose and tight interfaces. An approximate twofold axis perpendicular to the cell membrane relates the pair of P110–P140 heterodimers of a Nap. Fitting of the P140 and P110 subunits is unambiguous. The interface between the P110 and P140 subunits, defined as tight in the cryo-EM structure of the Nap, is the same found in the cryo-EM structure of the P140–P110 heterodimer. The interfering loop and the binding site are indicated with a black star. **c** The lateral view along the loose interface shows the V-like shape adopted by the two P140 subunits. **d–f** Three views of the cryo-ET map of the Nap, with the structures of P140 and P110 accurately fitted. The membrane bilayer of the mycoplasma cell is clearly defined in the lower part of the lateral views (**d**, **f**). **e** The top view shows that the tight interface is wider in the cryo-ET "open" conformation than in the cryo-EM "closed" conformation of the Nap. **f** The lateral view along the loose interface shows an interaction between heterodimers.

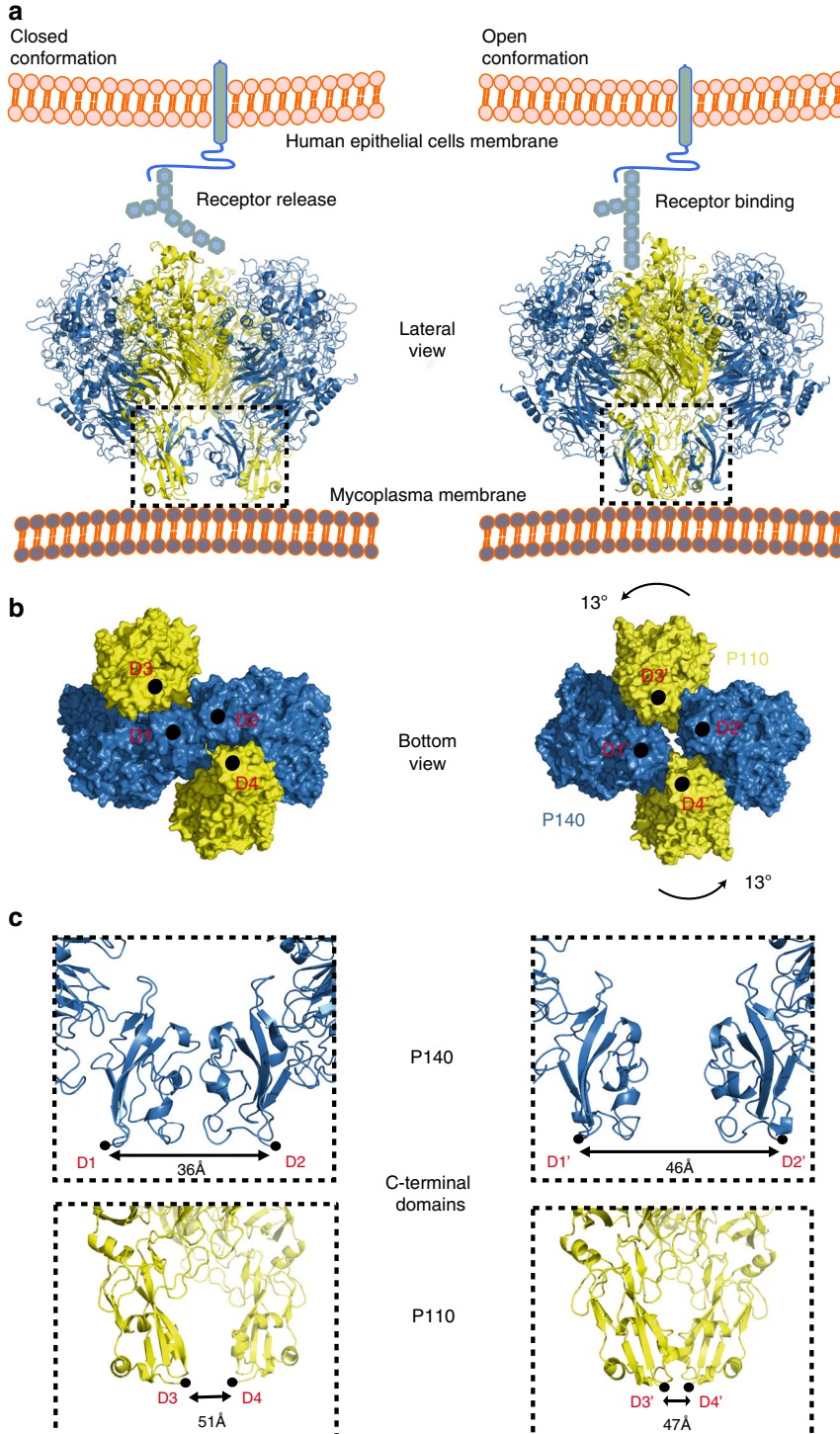

**Fig. 4 Structural rearrangements between "open" and "closed" conformations. a** "Closed" (left panel) and "open" (right panel) conformations of the Nap complex. The "closed" conformation associated with the release state of the sialylated oligosaccharides cell receptors. The "open" conformation associates with the bound and ready-to-bind state. **b** Bottom view along the Nap twofold axis, depicting points of contact (black circles) between subunits and the mycoplasma cell outer membrane. These points also mark the start (the N-terminal ends) of the transmembrane helices. The N-terminal end positions are changing from a squared to rhomboid shape, when transitioning from the closed to the open conformation. **c** Changes to the relative positions of the C-terminal domains from P140 (upper panels) and P110 (lower panels).

implies that between the cryo-ET and cryo-EM structures there is a rotation of P110 relative to P140 in the heterodimer and an increase of 10 Å in the closest distance between the C-terminal domains of P140 subunits (Fig. 4b, c, Supplementary Fig. 12, Supplementary Movies 3 and 4). The loose interface between

P140 and P110 subunits, which is very wide in the cryo-EM structure of the Nap, is narrower in the cryo-ET structure, with a few interactions between P140 and P110 (in particular, P140 residues 1175–1179 interacting with the P110 loop 750–755) that are absent in the cryo-EM Nap (Fig. 3c, f). The tight interface

between P140 and P110, which is in the "closed" conformation in the cryo-EM structure of the Nap, undergoes major rearrangements in the cryo-ET structure, becoming on average ~7 Å wider (Fig. 3a, b, d, e). This widening "opens" the sialylated oligosaccharide-binding site, making it accessible for binding in the cryo-ET structure of the Nap. Therefore, the cryo-ET and cryo-EM structures of the Nap correspond to conformations that, with respect to the sialylated host cell receptors, are respectively "open", with the binding site accessible and ready for binding, and "closed", with the binding site occluded and not accessible for binding. In the cryo-ET map, the cell membrane surrounding the Nap is visible as a double layer that is perpendicular to the Nap twofold axis. The positions of the C-terminal domains are largely different between the cryo-ET and cryo-EM Nap structures, changing from a squared to rhomboid shape, respectively (Fig. 4b). These differences also affect the positions of the transmembrane helices that follow in sequence immediately after the C-terminal domains, which provide a mechanism for communication between the Nap extracellular and intracellular regions. On the intracellular side, the cryo-ET map shows a compact density, similar to the one reported previously[3], with a volume suited to about 400 residues and an approximate twofold symmetry, where the four subunits appear to interact closely as they cannot be individualized at the resolution available.

## Discussion

Integration of the information obtained from the structural approaches shows that the Nap is composed of two P140–P110 heterodimers with a loose interface between them in the extracellular region, suggesting that subunits in the Nap are held together mainly by interactions in the intracellular and transmembrane regions. Each heterodimer of the Nap appears to undergo large structural rearrangements between the "open" and "closed" conformations, which are associated with attachment and release to cell receptors, respectively (Fig. 4, Supplementary Fig. 12, Supplementary Movie 4). The "open" conformation corresponds to a state in which the P140 and P110 extracellular regions from the four subunits of the Nap interact only weakly with each other, allowing P110 to remain in a conformation that is ready to bind or is bound to sialylated cell receptors. The "closed" conformation corresponds to a state in which the tight interaction of the extracellular regions of P140 and P110 occlude the cell receptor-binding site. The "closed" conformation could occur rapidly to release the Nap from the sialylated cell receptor. To avoid been trapped in the overall most stable state, the cycling between "open" and "closed" conformations would require a net input of energy in each Nap complex. The structural rearrangements in the Nap extracellular region can be transmitted to the cell interior through repositioning of the transmembrane helices. In vivo variants of Nap complexes indicate that structural rearrangements can play a critical role in both cell adhesion and motility. Results from this work, together with previous data, could form the basis for developing therapeutic strategies targeting the adhesion machinery of mycoplasmas from the *pneumoniae* cluster.

## Methods

**Cloning, expression, and purification of P110N, P140, and P110**. The region corresponding to the MG_192 gene from *M. genitalium* (strain G37, residues 23–827, P110N) was amplified from a synthetic clone (Supplementary Table 6) using primers P110N-F and P110N-R as Forward and Reverse, respectively (Supplementary Table 7). The PCR fragment was cloned into the expression vector pOPINE[20] (gift from Ray Owens plasmid #26043, Addgene, Watertown, USA) to generate a C-end His$_6$-tagged protein. The recombinant protein was obtained after expression in B834 (DE3) cells (Merck, Darmstadt, Germany) at 20 °C o/n upon induction with 1 mM IPTG at 0.6 OD$_{600}$. Cells were harvested and lysed in 1× phosphate-buffered saline (PBS) buffer by sonication. Subsequently, the cell extract

was centrifuged at $25,000 \times g$ at 4 °C and the supernatant applied to a 5 ml Histrap column (GE Healthcare, Pittsburgh, USA) equilibrated with 1× PBS as binding buffer and 1× PBS with 500 mM imidazole as elution buffer. Soluble aliquots of His$_6$-tagged P110N were pooled and loaded onto a HiLoad Superdex 200 16/60 column (GE Healthcare) in buffer consisting of 50 mM Tris pH 7.4 and 150 mM NaCl. The region corresponding to the MG_192 gene from *M. genitalium* (strain G37, residues 23–938, P110) was amplified from a synthetic clone (Supplementary Table 6) using primers P110-F and P110-R as Forward and Reverse, respectively (Supplementary Table 4). The region corresponding to the MG_191 gene from *M. genitalium* (strain G37, residues 23–1351) was amplified from a synthetic clone (Supplementary Table 6) using primers P140F and P140R as Forward and Reverse, respectively (Supplementary Table 4). Protein production and purification for P140 and P110 followed the same protocol as for P110N[6].

**Preparation of P140–P110 and P140–P110N complexes and SEC-MALS analysis**. Central fractions of the Superdex 200 16/60 column (GE Healthcare) from P140 and either P110N or P110 were mixed in a 1:1 ratio to a final concentration of ~11 mg/ml (measured with an absorption coefficient of 1.32). Each mixture was incubated for 30 min at room temperature to obtain the heterodimeric complex. For protein oligomerization experiments, prior to sample injections, the multi-angle light scattering (MALS, WYATT Technologies & Corporation) detector was normalized with 25 µl of BSA at 5 mg/ml. Then 150 µl of each purified P140, P110, and P140–P110 sample at 0.6, 2, and 0.5 mg/ml, respectively, were injected at 0.5 ml/min using a Superdex 200 increase 10/300 GL column (GE Healthcare) equilibrated with a running buffer consisting of 50 mM Tris pH 7.4 and 150 mM NaCl. All samples were purified by filtering through 0.22 µm filters.

**Preparation of purified Nap complexes**. A P110His strain was generated for purification of the Nap complexes from *M. genitalium* G37 cells (ATCC 33530). Produced by genetic engineering[3], the strain carries a 6xHistag insertion in the MG192 gene. Four liters of the P110His strain grown in SP4 medium in suspension at 37 °C at 150 r.p.m. was harvested by centrifugation ($16,000 \times g$, 30 min). The pellet was washed twice with PBS with calcium and magnesium, followed by cell disruption by sonication in PBS in the presence of 1 mM EDTA, 5 mM β-mercaptoethanol, 0.1 mM PMSF, and EDTA-free cocktail of protease inhibitors (Roche Diagnostics, Mannheim, Germany). The pellet generated after centrifugation ($70,000 \times g$) was resuspended in 75 mM Tris pH 7.4, 400 mM NaCl, 5% glycerol, and 2% n-octyl-β-D-glucopyranoside detergent by homogenization in a glass homogenizer. Solubilization of membranes was done overnight at 4 °C in an orbital shaker. Solubilized membranes were centrifuged at $50,000 \times g$, 30 min (4 °C) and the supernatant was purified by Ni$^{2+}$-affinity chromatography in 75 mM Tris pH 7.4, 400 mM NaCl, 5% glycerol, and 0.5% octylglucoside detergent. The purified Nap complex was obtained by Superose 6 size-exclusion chromatography equilibrated with the same buffer.

**Crystallization of P140 alone and the P140–P110N complex**. Screening for initial crystallization conditions was performed with 150 nl droplets in 96-well plates on a Cartesian robot (Cartesian (TM) Dispensing Systems) for both P140 alone and the P140–P110N complex. Optimized crystals from P140 were prepared by mixing 1 µl P140 at 6.5 mg/ml and 1 µl reservoir solution containing 20% PEG3350, 0.1 M BisTris Propane pH 7.5, and 0.2 M sodium sulfate hydrate at 20 °C in hanging drop in 24-well plates. Crystals from the P140–P110N complex were prepared by mixing 1 µl of P140–P110N complex at 10.95 mg/ml and 1 µl reservoir solution containing 23% PEG 500MME, 4% PGA, and 0.1 M sodium cacodylate pH 6.5 at 20 °C in hanging drop in 24-well plates. All crystals were flash-cooled in liquid nitrogen with 20% glycerol as a cryo-protectant.

**X-ray data collection and structure determination**. X-ray diffraction experiments were performed at the Xaloc Beamline (ALBA, Spain). Data were processed with Xia2[21] using XDS[22], Aimless and Pointless[23] from the CCP4i suite of programs[24]. P140 crystals belong to the space group C2 with six subunits in the asymmetric unit, while P140–P110N crystals belong to the space group P2$_1$ and contain four heterodimers in the crystal asymmetric unit. For the P140–P110N crystals, a partial molecular replacement solution was obtained with Phaser software[25] using the N-terminal domain of the available structures of P110 (PDB code 6RT3) as a search model. In these P140–P110N crystals, an initial mask was tentatively defined for P140 using the sub-tomogram averaged map of a whole Nap complex obtained from cryo-ET images of *M. genitalium* ghost cells. Density modification in the crystal of the P140–P110N complex, by iterative non-crystallographic symmetry averaging and solvent flattening followed by phase extension with DM[26], allowed us to obtain a rough density for P140, in which a few secondary structural elements were recognizable. This density was sufficient to provide an initial molecular replacement solution for the P140 crystals (with six subunits in the asymmetric unit) and for crystals of the orthologous protein P1 from *M. pneumoniae* (now deposited in the PDB with code 6RC9). Averaging within and between crystals, while updating and refining the masks, provided electron density maps from which it was possible to progressively build in parallel the structures of P140 and P1. The weak selenium methionine anomalous data, available from both the P140 and the P140–P110N crystals, were used to confirm

the sequence assignment to the P140 structure. Final models were traced with Coot[27] and refined with Refmac5[28] (Supplementary Table 1).

**Cryo-ET and sub-tomogram averaging**. Ghost cells were prepared in a similar manner as reported previously[3]. Briefly, cells of the adherently growing *M. genitalium* G37 strain (ATCC 33530) were cultivated on 300-mesh lacey carbon EM grid (PLANO GmbH, Wet zlar, No. S166-3) in 3 ml SP4 medium, rinsed for 1 min with PBS before incubation with 20 mM TEA pH 7.5, 0.5 M KCl, and 1% (v/v) Triton X-100 for 1–5 min in the Vitrobot Mark IV (Thermo Fisher Scientific, Waltham, USA) humidity chamber at 95% humidity and 15 °C. Before plunge freezing, 5 nm gold fiducial markers (Protein A-conjugated to gold beads, Cell Biology Department, UMC Utrecht) were added. Tilt-series were recorded using SerialEM 3.5.8[29] on a 300 kV Titan Krios (Thermo Fisher Scientific) in energy-filtered transmission electron microscopy (EFTEM) mode using a GIF Quantum® SE post-column energy filter (Gatan, Inc., California, USA) set to 20 eV slit width. A K2 Summit detector (Gatan Inc.) was operated in counting mode with a dose rate of ~1.0$e^-$/pixel/s, using a frame rate of 0.25 s and a total dose in the range of 40–50$e^-$/$Å^2$. A nominal magnification of ×64,000 was used with the K2 in super-resolution mode, resulting in a pixel size of 1.1 Å/pixel. The tilt-series generally covered an angular range from −60° to +60° with an angular increment of 3° and defocus set between −2 and −4 μm using either dose symmetric or bidirectional acquisition. Tomograms were reconstructed by super-sampling SART[30] with 3D contrast transfer function (CTF) correction. For volume visualization and isosurface rendering, the EMpackage in Amira was used (Thermo Fisher Scientific)[31]. Sub-tomogram averaging was performed in a similar manner as reported previously[3]. In short, Nap complexes were manually selected and pre-aligned according to their orientation on the membrane, which provided a strong constraint lowering the degrees of freedom for sub-tomogram averaging. This first pre-aligned average was used as the starting reference template for the iterative sub-tomogram averaging procedure, thus completely avoiding any initial model bias. The "open" structure for the cryo-ET data is maintained when using the "closed" structure as a starting reference (Supplementary Fig. 13). We then averaged 11,000 Nap sub-tomograms. Post-processing involved refinement by projection matching and dose weighting of the wedge components. After classification, 8800 Nap sub-tomograms were included in the final average. All processing was performed using sub-tomograms with a voxel size of 0.44 nm. The 0.5 criterion was used for resolution estimation. For classification of sub-tomograms, the averaged particles were normalized, and single projection slices in *X*, *Y*, and *Z* directions were extracted using custom MATLAB scripts, which are available on GitHub (https://github.com/uermel/Artiatomi). Subsequently, the two-dimensional (2D) projection slices were imported into Relion[32] and 2D classification without image alignment was performed. Constituent sub-tomograms from the respective 2D classes were summed together to create 3D classes. Molecular graphics rendering and analyses were performed with either UCSF Chimera[16] or UCSF ChimeraX[33] (Resource for Biocomputing, Visualization, and Informatics at the University of California, San Francisco, USA). In all cases, rigid-body transformations were used to fit X-ray structures into the cryo-EM density maps using the fit-in-map function in UCSF Chimera.

**Single-particle cryo-EM**. For single-particle cryo-EM, a 3.5 µl aliquot of 20 µg/ml purified P140–P110 heterodimer, or 25 µg/ml purified Nap complex, was applied to a (30 s) glow-discharged R1.2/1.3 Quantifoil grid (Quantifoil, Großlöbichau, Germany), and plunge-frozen in liquid ethane (Vitrobot Mark IV, Thermo Fisher Scientific) at 100% humidity, 4 °C, blot force −3, wait time 60 s, with a blotting time of 15 s for the heterodimer and 20 s for the Nap complex. Dose-fractionated movies were collected at a nominal magnification of ×130,000 (1.05 Å per pixel) in nanoprobe EFTEM mode at 300 kV with a Titan Krios (Thermo Fisher Scientific) electron microscope using a GIF Quantum S.E. post-column energy filter in zero loss peak mode and a K2 Summit detector (Gatan Inc.). For the heterodimer, 32 frames of 0.2 s each were collected and for the Nap complex 34 frames per micrograph were collected. The camera was operated in dose-fractionation counting mode with a dose rate of ~7.7 electrons per $Å^2$ $s^{-1}$ for the heterodimer and ~7.4 electrons per $Å^2$ $s^{-1}$ for the Nap complex, resulting in both cases in a total dose of ~50 electrons per $Å^2$. Defocus values ranged from −1 to −4 µm with marginal (<0.1 µm) astigmatism.

**Single-particle cryo-EM image processing**. Relion 3.0 was used for the whole-image processing workflow[32] unless stated otherwise. Beam-induced motion correction was performed using UCSF MotionCor2[34]. The CTF of each micrograph was estimated using GCTF[35]. As reference for autopicking the heterodimer, the crystal structure filtered to 20 Å was used. As a reference for the Nap complex, the crystal structure of the dimer was fitted into the cryo-ET data and subsequently lowpass filtered to 20 Å. Cross-checking experiments with Laplacian-of-Gaussian autopicking were performed for the Nap to verify that the structure was not reference-biased. For the heterodimer, 1,024,402 particles were extracted from 2751 micrographs, and for the Nap complex, 160,203 particles were extracted from 2225 micrographs, using a 270 pixel box in both cases. All images were normalized to make the average intensity of the background equal to zero during pre-processing. False-positive particles were removed manually or by unsupervised 2D classification. The CTF-refinement function from Relion 3.0 was used to perform per-

particle defocus estimation using a search range from 300 nm around the estimated values from the whole micrographs. To correct for local motion and for radiation damage, we used the Bayesian polishing function of Relion 3.0, in which the resolution-dependent decay caused by radiation damage is taken into account[36]. The polished particles were subjected to another round of 2D classification to remove remaining junk particles. The remaining 404,064 particles for the heterodimer and 28,000 particles for the Nap complex were used for 3D classification using the initially generated density map filtered to 60 Å as reference. For the heterodimer, six classes were generated using 3D classification, of which one class (37,009 particles) showed electron density in the stalk region for P110 and was refined using 3D auto-refine. For the Nap complex, six classes were generated using 3D classification, of which two classes (11,737 particles) were tetrameric. The one tetrameric class containing 5612 particles was refined using 3D auto-refine. Gold-standard Fourier shell correlations (FSCs) were calculated during the 3D refinement between two independently refined halves of the data. The resulting map was post-processed to exclude solvent regions from the FSC calculation and to perform sharpening with a temperature factor of −78 $Å^2$ for the heterodimer and −218 $Å^2$ for the Nap complex resulting in a global resolution of 4.1 Å for the heterodimer and 9.8 Å for the Nap complex, using the 0.143 criterion. Local resolution was estimated using Relion 3.0, showing high resolution in the core regions of the dimer and low resolution in the surface and stalk regions (Supplementary Figs. 8 and 9).

**Surface plasmon resonance**. For binding assay experiments, a Biacore 3000 biosensor platform (GE Biosystems) equipped with a research-grade streptavidin-coated biosensor chip SA was used. The chip was docked into the instrument and preconditioned with three 1-min injections of 1 M NaCl in 50 mM NaOH. Both 3SL-PAA-biotin and 6SL-PAA-biotin (Carbosynth) oligosaccharides were injected over the second and third flow cell, respectively, at 10 µg/ml diluted in HBS-P (10 mM Hepes, pH 7.4, 0.15 M NaCl, and 0.005% P20). The first cell was left blank to serve as a reference. The running buffer consisted of HBS-P at a flow rate of 30 µl/min and the immobilization levels acquired were ~160 and ~180 response units for 3SL-PAA-biotin and 6SL-PAA-biotin, respectively. A series of diluted purified extracellular P140 and P140–P110 samples in HBS-P (1.25, 2.5, 5, 10, and 20 µM) were injected over the flow cell surface at 30 µl/min. Interaction analysis were performed at 25 °C and the protein was allowed to associate and dissociate for 60 and 90 s, respectively, followed by a 30 s regeneration injection step of 0.05% SDS at 30 µl/min.

**Strains, culture conditions, and primers**. *M. genitalium* was grown in SP4 medium (Hardy Diagnostics) at 37 °C in tissue culture flasks. Mutants were isolated on SP4 agar plates supplemented with puromycin (3 µg/ml). All *M. genitalium* strains used in this work are listed in Supplementary Table 4. *Escherichia coli* XL1-Blue strain (Agilent, Santa Clara, USA) was used for cloning and plasmid amplification purposes. It was grown in LB broth or on LB-agar plates containing 100 µg/ml ampicillin. Primers used are listed in Supplementary Table 7.

**DNA manipulation and mutant construction**. Plasmid DNA was purified using GeneJET Plasmid Miniprep Kit (Thermo Fisher Scientific). PCR products and DNA fragments were recovered from agarose gels using NucleoSpin Gel and PCR Clean-up Kit (Macherey-Nagel, Düren, Germany), and digested using the corresponding restriction enzymes (Thermo Fisher Scientific) when necessary. For transformation of *M. genitalium*, plasmids were purified using the GenElute HP Midiprep Kit (Sigma-Aldrich, St. Louis, USA) following the manufacturer's instructions.

P110 variants were generated in a two-step PCR procedure using DNA from plasmid pTnPacP110-WT as a template[6]. For each mutant, the first PCR round was performed using primer COMmg192-F and the specific mg192 reverse primer, or primer COMmg192-R and the specific mg192 forward primer (Supplementary Table 4). To reconstitute the mutated full-length MG_192 alleles, we conducted splicing by overhang extension (SOE) PCR using the specific amplicon pair obtained for each mutant as a template (R600A, Y830A, R834G, D836L, W838F, and G839F) and primers COMmg192-F and COMmg192-R. Then, the mutated P110 alleles were digested with *Apa*I and *Xho*I and ligated into a similarly digested pMTnPac plasmid backbone[6] to generate the corresponding pMTnPacP110 plasmid series. The P110 variant carrying the triple substitution RQD was obtained using primers Q460AD461Amg192-F and -R to introduce the double substitution Q460A and D461A into the pMTnPacP110-R600A plasmid, which was used as the DNA template for the PCR reaction. Sequencing analysis of the different TnPacP110 minitransposons, using primers Tnp3, RTPCR192-F, RTPCR192-R, and PacUp, ruled out the presence of additional mutations in the MG_192 sequence. These plasmids were transformed into a *M. genitalium* MG_192 null mutant to create the different P110 variant strains. Identification of the minitransposon insertion site in the individual clones was done by sequencing using the PacDown primer and chromosomal DNA as a template.

**Transformation and screening**. *M. genitalium* MG_192 null mutants were transformed by electroporation using 5 µg of plasmid DNA of the different minitransposons, as previously described[37]. Puromycin-resistant colonies were

picked, propagated, and stored at –80 °C. For screening purposes, strains were further propagated in 25 cm² tissue culture flasks with puromycin and lysed using 0.1 M Tris-HCl pH 8.5, 0.05% Tween-20, and 250 μg/ml Proteinase K for 1 h at 37 °C. Then, Proteinase K was inactivated at 95 °C for 10 min. *M. genitalium* lysates were screened by sequencing using the PacDown primer. By contrast, the MG_192 alleles were fully re-sequenced to rule out the presence of undesired mutations.

**Sequencing reactions**. Sequencing reactions were performed with the BigDye® v3.1 Cycle Sequencing kit using 2.5 μl of genomic DNA, following the manufacturer's instructions (Thermo Fisher Scientific). All sequencing reactions were analyzed using an ABI PRISM 3130xl Genetic Analyser at the Servei de Genòmica i Bioinfòrmatica (UAB).

**SDS-PAGE**. Whole-cell lysates were obtained from mid-log phase cultures grown in 75 cm² flasks. Protein concentration was determined with the Pierce™ BCA Protein Assay Kit (Thermo Fisher Scientific), and similar amounts of total protein were separated by SDS-PAGE following the standard procedures.

**Quantitative hemadsorption assay**. We used $10^9$ mycoplasma cells during the hemadsorption assay. Fluorescence-activated cell sorting (FACS) data were acquired using a FACSCalibur (Becton Dickinson, Franklin Lakes, USA) equipped with an air-cooled 488 nm argon laser and a 633 nm red diode laser and analyzed with the CellQuest-Pro and FACSDiva software (Becton Dickinson). Hemadsorption was quantified using flow cytometry as previously described[38] with few modifications. Binding of mycoplasma cells to red blood cells can be modeled in an inverse Langmuir isothermal kinetic function:

$$M_f = 1 - \frac{B_{max}[RBC]}{K_d + [RBC]}. \qquad (1)$$

The plots represent the best-fitting curves to a series of hemadsorption measurements obtained from at least two biological repeats for each strain. We performed a double-gating strategy, using a preliminary FL3-H/FL2-H gate following an SSC-H/FL1-H gate, as described previsouly[38].

**Time-lapse microcinematography**. Gliding properties of the wild type and mutant strains were analyzed by time-lapse microcinematography as previously described[5]. Samples from mid-log phase cell cultures were grown overnight on eight-well ibiTreat μ-slides (Ibidi, Gräfelfing, Germany) filled with 200 μl of SP4 medium. Prior to the observation, medium was replaced with fresh SP4 pre-warmed at 37 °C and cell movement was examined at 37 °C and 5% $CO_2$ using a Nikon Eclipse TE 2000-E inverted microscope equipped with a $CO_2$ Microscope Cage Incubation System (Okolab, San Bruno, USA). Images were captured at 2 s intervals for a total of 2 min for all different strains. The frequency of motile cells was determined by examining approximately 250 isolated cells of each strain. Given that cells of the P110-W838F mutant were mainly aggregated, only 64 individual cells could be analyzed. Mean velocities were obtained from the analysis of at least 25 motile, isolated cells, by measuring the traveled distance and dividing this value by the time spent in microcinematography. Analysis of the gliding properties was performed using the ImageJ software with the MTrack2, MTrackJ, wrMTrack, and Stack Deflicker plugins (https://imagej.nih.gov/ij/).

**Reporting summary**. Further information on research design is available in the Nature Research Reporting Summary linked to this article.

## Data availability
Atomic coordinates and structure factors for the reported crystal structures of P140 and the P140–P110N complex have been deposited in the Protein Data Bank under accession codes 6RUT and 6S3U, respectively. The cryo-electron microscopy density for the 4.1 Å density map of the heterodimer was deposited in the Electron Microscopy Data Bank (EMDB) under the accession code EMD-10890 and the fitted P140–P110N complex under the accession code 6YRK. The 9.8 Å density map of the single-particle cryo-EM density map of the Nap was deposited in the EMDB under the accession codes EMD-10260. The 15 Å in situ density map of the in situ cryo-ET Nap was deposited in the EMDB under the accession code EMD-10259. The source data underlying Fig. 2g are provided as a Source Data file. Other data are available from the corresponding authors upon reasonable request.

## Code availability
The reconstruction, sub-tomogram averaging, and classification code referenced in the methods is available:
(i) Artiatomi: http://github.com/uermel/Artiatomi
(ii) Classification: https://github.com/lsprankel/Aparicio_2020.

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

## Acknowledgements
We thank Utz Ermel for providing custom Matlab scripts. This work was supported by grants BFU2018-101265-B-100 and BIO2017-84166-R from MINECO (Spain). A.S.F. acknowledges the Deutsche Forschungsgemeinschaft Grant FR1653/6-1 and SFB902, as well as the Loewe Dynamem funding for the project. D.A. acknowledges María de Maeztu Unit of Excellence grant MDM-2014-0435. Many thanks are given to the XALOC beamline team at ALBA for all the support during data collection and, in particular, to Dr. Barbara M. Calisto for her many contributions to the mycoplasma structural studies. Many thanks are also given to the Crystallography Platform at the Barcelona Science Park (PCB).

## Author contributions
Conceived and designed the experiments: D.A., M.P.S., J.P., O.Q.P., I.F., and A.S.F. Performed the experiments: D.A., M.P.S., M.M.-S., D.V., M.R., M.S.W., A.S., J.R., L.S., S.T.-P., L.G.-G., and O.Q.P. Analyzed the data: D.A., M.P.S., M.M.-S., D.V., M.S.W., J.R., L.S., J.P., O.Q.P., I.F., and A.S.F. Contributed reagents/materials/analysis tools: E.Q., J.P., I.F., and A.S.F. Wrote the paper: D.A., M.S., O.Q.P., I.F., and A.S.F.

## Competing interests
The authors declare no competing interests.
