## [Peer Review File · Nature Communications]

Reviewers' comments:

Reviewer #1 (Remarks to the Author):

This paper clarified the atomic level structures of P140 and a complex with P110, which is the most exciting subject in mycoplasmaology. The structures are consistent with the previous information and the story is straightforward. However, the paper is sometimes confusing and misleading. Major points are listed below and minor ones are commented in a pdf.

Major points

1) In my understanding, this study used P110N and P140 for crystal, and P110 and P140 for EM of dimer and tetramer, and FL P110 and FL P140 for EM of native Nap. This is not easy to trace. Please clarify.

2) L34 and many positions: Distinguish reconstituted P110-P140 tetramer without cytoplasmic domain and the Nap structure recovered from a cell.

3) The structural similarity between P110 and P140 is a discovery and should be stressed. Please present it by a figure.

4) L85-: The interfering loop is a major finding in this paper. However, the authors cannot conclude if it happens only in the artificial P110-P140 complexes or it has a special role in the control of binding, from mutant experiments. I am curious about the activity of P110-RQD mutant cells. Even if the protein amount is small, some specific effects can be detected on binding and gliding.

5) L100-: Can you explain briefly again the reason why you focus on potassium binding?

6) L113-: "Cells of the P110 mutants Y830A, R834G, and D836L also exhibited altered gliding velocities, indicating that structural integrity at the interface between the N- and C-domains, away from the cell receptor binding site, is critical for motility in *M. genitalium*." As the gliding speeds of mycoplasmas are affected by many conditions, 20% difference may not be a significant change. However, it is author's judge. If the authors want to claim it, they should show the results of "test".

7) L130-: Can you show us briefly again the preparing method of the tetramer?

8) In my understanding, the artificial hexamer does not contain the cytoplasmic domain but the Nap structure from cells contains it. If this is correct, please discuss about it. The "closed form of tight interface" may be stimulated by the lack of cytoplasmic domain, even if the interfering loop has physiological roles in binding and gliding.

Reviewer #2 (Remarks to the Author):

Aparicio et al. present a study of the *M. genitalium* Nap adhesion complex. A combination of X-ray crystallography, single-particle cryo-EM and cryo-tomography is used to understand the structure of Nap components and of the assembled complex in vitro and in situ. Functional assays are also used to analyse structure-derived mutants.

The *M. genitalium* Nap complex is an important target, and structural information is still lacking. The work presented here is therefore important and timely. Taking an approach that combines many structural techniques at complementary levels of resolution and physiological relevance is powerful, and allows to obtain a comprehensive understanding of the system studied. Here the

authors interpret their structures to propose an open-close mechanisms that would sense the presence of a sialylated ligand and transmit a signal to the intracellular environment, which is a potentially important conclusion.

However, there are a number of issues that I feel need to be addressed before this work can be published:

Major comments:

The subtomogram averaged structure presents a looser tight interface and tighter loose interface, i.e. it becomes more '4-fold symmetric' than the SPR structure.

Can the authors exclude this is an artefact of misalignment?

They should try and align the data against a starting reference derived from the SPR structure and show they still get to the same result.

In the main text (lines 156-160) they mention classification, and a 'most abundant class'. They then refer to the open state as the most abundant class and speculate the closed state is the less abundant (lines 192-197). But how does the minor class look like? Can they interpret it to the point of assigning it to the closed conformation? If so, this needs to be shown in a figure. If not, their conclusion that the closed conformation is present in situ and is a minority conformation becomes speculative and should be removed.

Also, in the methods there is no mention of classification in the subtomogram averaging section. This needs to be included.

Along similar lines: how do they know the SPR structure (closed) has any physiological relevance? They should include in their discussion the possibility that the SPR structure is forced into a different conformation during the purification procedure. Is there any previous evidence that the Nap switches between conformation? (I am not knowledgeable in this biological field). If so, this should be cited and discussed in support of their conclusion.

Minor comments:

line 82: RMSD between what? I assume it is alpha-chain since the sequences are different, but this needs to be specified.

line 91: can the SPR results be shown as a supplementary figure?

figure 2, panel f: it is not clear from the figure how the region shown participates to the heterodimer interface. Can the panel be modified so that it becomes clear how 'f' relates to the overview in panel 'e' and where the interface is.

figure 2, panels a and e: it looks from this view that the two portions that are present either in the Xray or single particle structures would clash if superimposed. Is this the case? If so this needs to be explained. If not, showing the panels from a different angle would help.

Reviewer #3 (Remarks to the Author):

The manuscript describes a structural characterisation of the P140:P110 complex from *M. genitalium* by X-ray crystallography and cryoEM. It represents an extension of previous work on the complex by the same group using tomography (doi.org/10.1111/mmi.13743) and the crystal structure of P110 ([doi:10.1038/s41467-018-06963-y](https://doi.org/10.1038/s41467-018-06963-y)). The novel aspects are the crystal structures of P140 alone and in complex with P110N, some mutagenesis work to validate the occlusion of the sialic acid binding site in the complex and reconstructions from cryoEM and cryoET of the complex at lower resolution. The authors provide evidence that the sialic acid binding site is blocked in the P140:P110N crystal structure. Interaction between the two proteins is important for motility and,

presumably, sialic acid (SA) binding (through hemadsorption): mutation of key contact residues for P140 on P110 changed hemadsorption phenotypes and altered motility. The crystallographic parts of the project were clearly challenging and the authors should be congratulated on resolving these difficult structures.

The main conclusion of the manuscript- implied by the title- is that the Nap complex adopts 'open' and 'closed' conformational states *in vivo* which alternate and therefore modulate SA-mediated binding. Critically, this conclusion is based on a comparison of structures derived from single particle cryo-EM, from the *in vitro* reconstituted complex, and the *in situ* cryo-ET reconstruction (Fig 3). The detergent-solubilised Nap complex adopts a proposed 'closed' structure which is not competent for SA binding but the *in situ* cryo-ET density map, obtained at lower resolution, appears to show an 'open' conformation. We would presume that the latter structure is also competent for SA binding. The difficulty is that the authors then propose that both conformational states exist *in vivo* and alternate in some way, perhaps associated with some form of trans-membrane signalling (lines 197-200). This is highly speculative and the argument has an obvious flaw. What are the data to establish that the 'closed' conformation is not an artefact of detergent extraction or, in the case of the crystal structure, formation of the complex *in vitro* between the two recombinant proteins? Put another way, what is the evidence that the 'closed' state exists *in vivo* at all? There seems to be little direct evidence from the cryoET data.

A second area which needs clarification are the methods used for fitting models into cryo-EM density maps. This is important for the case which the authors are making but few details are provided. It is particularly important for the fitting into the lower resolution maps shown in Fig 3. How was the fitting carried out- methods and relevant parameters (eg correlation coefficients) should be supplied? What is the evidence that the fitted models are unambiguous, unique solutions? Given the similarities between the P140 and P110 structures (lines 77-78), is it conceivable that the positions of each protein in the complex could be swapped? An argument based on a more thorough search of possible models, followed by refinement, is needed.

I would recommend that the authors reprocess their SPA without using the crystal structure as a reference (line 322). Even when filtered at 20Å resolution, this can introduce bias. Using G-auto with a Gaussian blob or Warp will pick particles without bias and high degree of accuracy. Reference-free classes can be generated using a small scale hand pick and then used as a reference.

Minor points

Line 78 The authors do not elaborate on the observation that the P140 and P110 proteins share similar fold topologies. A figure to illustrate this observation would be useful.

Line 85 Have the authors analysed the interface between P140 and P110 using the PISA protein interface server, to verify that this is the correct interface?

Lines 88-91 Where are these data? They should be included.

Line 120 (and elsewhere) state the resolution criteria used.

Lines 169-721 The binding site is not marked in Fig 3a,d, so this point is hard for the reader to visualise.

Supplementary Figure 5: it is not easy to tell from the figure, but the map does not look at be at 4.1Å resolution. Could the authors include some details of the density map and secondary structure elements to substantiate this?

Reviewer #4 (Remarks to the Author):

This study has continued the work from previously to obtain higher resolution structures of the adhesin proteins P110 and P140 by X-ray and single particle Cryo-EM. The authors also solved a

~10Å resolution single particle structure of a closed conformation of NAP solubilised in detergent and a ~15Å Cryo-tomography structure of the intact complexes in native membranes. Overall the study is well performed and the paper nicely written and makes use of multiple structural biology and biophysical techniques to address the conformational variability of NAP. At 4.1Å resolution the single particle reconstruction of the P110-P140 heterodimer should provide enough information to build a fairly accurate model which is mostly similar to the crystal structure. The ~10Å and 15Å structures however should be interpreted with more caution as the exact positions of loops/hairpins and individual secondary structure elements cannot be positioned with accuracy at this resolution range and the main text should reflect this limitation for the non-structural biologist reader. The authors could suggest the existence of an "open" conformation from the Cryo-ET map, as this cannot be unequivocally be stated with existing data.

For the two single particle structures the authors should show the following:

- Figures of a close-up density and model of the sialic acid binding site from different views to illustrate these two structures exist in a "closed" conformation.
- Explanation in the main text or methods on how the model was fitted and/if refined for the 10Å map.

For the Cryo-ET structure at 15Å resolution:

- In figure 3e there seems to be a wider space in the density of the tight interface but the model seems to be protruding through the density and the atomic contacts are not clear. Furthermore, in figure 3d there doesn't seem to be a gap in the density at the top/extracellular interface between P140 and P110. Again, a close-up figure of this region from various views should be shown to help re-enforce the authors conclusion/suggestion of an "open" state.
- The clash score of the fitted atomic models for all single particle Cryo-EM and Cryo-ET experiments should be shown in the supplement to illustrate that the fitted models represent are sterically feasible.

Minor points:

- Line 133. The resolution of the Cryo-EM map of NAP in detergent is listed as 9.8Å but the FSC reports 10Å and later in the methods (line 343) it is listed as 10.5Å.
- Line 587 refers to supplementary figure 4 but this should be supplementary figure 5?
- Line 127: In the main text the C-terminal domain of P110 is referred to as "well defined" but in Sup. Fig.5 it is referred to as low-resolution and flexible. Rather than "well defined" it should be referred to in the main text as clearly visible to be consistent and more accurate.
- Line 182: This density is not shown in any figures?

We would like to thank all reviewers for their very encouraging comments and constructive suggestions for our manuscript.

Reviewer #1:

This paper clarified the atomic level structures of P140 and a complex with P110, which is the most exciting subject in mycoplasmaology. The structures are consistent with the previous information and the story is straightforward. However, the paper is sometimes confusing and misleading. Major points are listed below and minor ones are commented in a pdf.

We thank the reviewer for the compliment. We have revised and edited the manuscript text in many places (edits can be seen in the related manuscript file which contains the tracked changes), which we hope improves the readability and coherence.

Major points

1) In my understanding, this study used P110N and P140 for crystal, and P110 and P140 for EM of dimer and tetramer, and FL P110 and FL P140 for EM of native Nap. This is not easy to trace. Please clarify.

The reviewer is correct in their understanding of the specimens used. Indeed, the EM tetramer also corresponds to the FL P110 and FL P140 as does the native Nap. The text has been modified in many positions related to this query in order to define accurately, in each case, the specimen(s) used.

2) I34 and many positions: Distinguish reconstituted P110-P140 tetramer without cytoplasmic domain and the Nap structure recovered from a cell.

We only observe tetramers when the full length P140 and P110 subunits (including the transmembrane helices and the cytoplasmic regions) are present.

We have changed the text to read: *“Next, we performed single-particle cryo-EM using a purified sample of Nap complexes, obtained as previously described³ (see Methods). The purified Nap complexes contain full-length P140 and P110 proteins, including the transmembrane helices and cytoplasmic regions, which are required for formation of tetramers.”*

3) The structural similarity between P110 and P140 is a discovery and should be stressed. Please present it by a figure.

We thank the reviewer for this suggestion.

A new supplementary figure has been added (the new supplementary figure 4) showing a side by side view of the overall structures of P110 and P140, where the similarities between the structures are emphasized.

4) I85-: The interfering loop is a major finding in this paper. However, the authors cannot conclude if it happens only in the artificial P110-P140 complexes or it has a special role in the control of binding, from mutant experiments. I am curious about the activity of P110-RQD mutant cells. Even if the protein amount is small, some specific effects can be detected on binding and gliding.

Adherence is a prerequisite for motility of mycoplasmas. Cells from the P110-RQD mutant are virtually non-motile. Characterization of the P110-W838F mutant, shown in Supplementary Table 2, underscores this statement. Unlike P110-W838F, cells from the P110-RQD mutant are completely unable to adhere to plastic surfaces, which prevents the analysis of this mutant by time lapse microcinematography.

As suggested by the reviewer, we assessed the binding activity of the P110-RQD mutant. Our assays indicate that mutagenesis of residues RQD abrogates the binding capacity of *M. genitalium* cells.

The text now reads: *Strains expressing the P110-RQD variant protein, which was barely detectable, showed a null binding capacity phenotype (Figure 2g and Supplementary Figure 6b). The variant protein P110-R600A was well expressed, but the strain presented no capacity for adherence and characterization of cell motility was not feasible.* “

5) line100-: Can you explain briefly again the reason why you focus on potassium binding?

We thank the reviewer for pointing this out: Mutations away from the sialic binding site altering adhesion and/or gliding could indicate conformational transitions during these processes. A first choice for mutations was the interfaces between N- and C-domains, where there are now experimental indications that structural flexibility can occur. The most distinctive feature in the vicinity of these interfaces is the potassium binding in P110. Therefore, we decided to focus, at least initially, on mutations close to the potassium site.

Figure 2F has been expanded and modified in this regard, and the manuscript text has been elaborated.

The text now reads: *“Five mutations were introduced close to the interface between the N-terminal and C-terminal domains, to check if either adhesion or motility was affected... The Y830A, R834G and D836L variants also exhibited altered gliding velocities, indicating that structural integrity at the interface between the N-terminal and C-terminal domains, away from the cell receptor binding site, is critical for motility in M. genitalium.”*

6) line113: “Cells of the P110 mutants Y830A, R834G, and D836L also exhibited altered gliding velocities, indicating that structural integrity at the interface between the N- and C-domains, away from the cell receptor binding site, is critical for motility in *M. genitalium*.” As the gliding speeds of mycoplasmas are affected by many conditions, 20% difference may not be a significant change. However, it is author’s judge. If the authors want to claim it, they should show the results of “test”.

We thank the reviewer for raising this important point regarding the statistical significance of the differences observed in gliding speed. In Supplementary Table 2, we now show the results of a Paired Student’s T-test. This analysis indicates that the reduction in motility observed in mutants P110-Y830A and P110-D836L is statistically significant. This is stated also in a footnote of the Supplementary Table 2.

7) L130: Can you show us briefly again the preparing method of the tetramer?

We include this new paragraph in the Methods section:

“Preparation of purified Nap complexes

*A P110His strain was generated for purification of the Naps from *M. genitalium* G37 cells (ATCC 33530). Produced by genetic engineering, the strain carries a 6xHistag insertion in the MG192 gene. Four liters of the P110His strain grown in SP4 medium in suspension at 37°C at 150 rpm was harvested by centrifugation (16 000 × g, 4°C, 30 min). The pellet was washed twice with phosphate buffered saline (PBS) with calcium and magnesium, followed by cell disruption by sonication in PBS in the presence of 1 mM EDTA, 5 mM β-mercaptoethanol, 0.1 mM PMSE, and EDTA-free cocktail of protease inhibitors (Roche Diagnostics Mannheim, Germany). The pellet generated after centrifugation (70 000 × g, 4°C) was resuspended in 75 mM Tris pH 7.4, 400 mM NaCl, 5% glycerol and 2% n-octyl-β-D-glucopyranoside detergent by homogenization in a glass homogenizer. Solubilization of membranes was done overnight at 4°C in an orbital shaker. Solubilized membranes were centrifuged at 50 000 × g, 30 min (4°C) and the supernatant was purified by Ni²⁺-affinity chromatography in 75 mM Tris pH 7.4, 400 mM NaCl, 5% glycerol and 0.5% Octylglucoside detergent. The purified Nap complex was obtained by Superose 6 size-exclusion chromatography equilibrated with the same buffer.”*

We also keep the reference to the publication where the method was used for the first time.

8) In my understanding, the artificial hexamer does not contain the cytoplasmic domain but the Nap structure from cells contains it. If this is correct, please discuss about it. The “closed form of tight interface” may be stimulated by the lack of cytoplasmic domain, even if the interfering loop has physiological roles in binding and gliding.

The purified Nap tetramers do contain the cytoplasmic domains from both P140 and P110. The text has been modified, as indicated in our reply to the reviewer queries 1 and 2, in order to clarify the properties of the different samples used.

Reviewer #2 (Remarks to the Author):

The subtomogram averaged structure presents a looser tight interface and tighter loose interface, i.e. it becomes more '4-fold symmetric' than the SPR structure.

Can the authors exclude this is an artefact of misalignment?

They should try and align the data against a starting reference derived from the SPR structure and show they still get to the same result.

We thank the reviewer for this comment. We can actually exclude a misalignment or reference bias. For the cryo-ET structure, only a membrane with a blob derived by the geometrical constraints of Nap being attached to the membrane was used. We have now also performed the experiment to show that when using the closed tetramer structure as it was elucidated in the crystal as a starting structure, that the sub-tomogram average indeed still converges to the open structure.

We have included a supplementary figure 13 showing the new experiment and have added in the methods section of the manuscript:

“In short, Nap complexes were manually selected and pre-aligned according to their orientation on the membrane, which provided a strong constraint lowering the degrees of freedom for sub-tomogram averaging. This first pre-aligned average was used as the starting reference template for the iterative sub-tomogram averaging procedure, thus completely avoiding any initial model bias. The open structure for the cryo-ET data is maintained when using the closed structure as a starting reference (Supplementary Figure 13).”

In the main text (lines 156-160) they mention classification, and a 'most abundant class'. They then refer to the open state as the most abundant class and speculate the closed state is the less abundant (lines 192-197). But how does the minor class look like? Can they interpret it to the point of assigning it to the closed conformation? If so, this needs to be shown in a figure. If not, their conclusion that the closed conformation is present in situ and is a minority conformation becomes speculative and should be removed.

We thank the reviewer for this comment. Two aspects need to be kept separate: (1) The classification of the sub-tomograms in order to exclude “bad” sub-tomograms in order to achieve a better resolution and (2) the classification of the sub-tomograms in those showing the open and those showing the closed conformation.

For the first point: We now present in Supplementary figure 11 the two classes, which are distinguished by a difference in the stalk region. For the final cryo-ET subtomogram average, we used the class where the stalks are well defined.

For the second point: Unfortunately, we do not see a closed-conformation in the cryo-ET classes, which lead us to the conclusion that this is a very fast/transient event, which is present in such a small minority of the population that classification cannot detect it.

We have now modified the discussion section in this regard: *“The “closed” conformation corresponds to a state in which the tight interaction of the extracellular regions of P140 and P110 occlude the cell receptor binding site. The “closed” conformation could occur rapidly to release the Nap from the sialylated cell receptor. To avoid been trapped in the overall most stable state, the cycling between “open” and “closed” conformations would require a net input of energy in each Nap complex.”*

Also, in the methods there is no mention of classification in the subtomogram averaging section. This needs to be included.

We thank the reviewer for pointing this out. We have now added these sentences in the manuscript under the methods section for cryo-ET:

“For classification of subtomograms, the averaged particles were normalized and single projection slices in X, Y and Z direction were extracted using custom MATLAB scripts which are available on GitHub (<https://github.com/uermel/Artiatomi>). Subsequently the 2D projection slices were imported into Relion and 2D classification without image alignment was performed. Constituent subtomograms from the respective 2D classes were summed together to create 3D classes.”

Along similar lines: how do they know the SPR structure (closed) has any physiological relevance? They should include in their discussion the possibility that the SPR structure is forced into a different conformation during the purification procedure. Is there any previous evidence that the Nap switches between conformation? (I am not knowledgeable in this biological field). If so, this should be cited and discussed in support of their conclusion.

To our knowledge, previous to the present structural results there were only indirect experimental evidences, based in recognition by monoclonal antibodies, about Naps experiencing conformational changes during gliding, however without any structural information (Seto, Kenri, Tomiyama & Miyata. J.Bacteriology 2005, 187: 1875-7). To our knowledge this is the first study to provide the structural basis for the control mechanism of adhesion.

Minor comments:

line 82: RMSD between what? I assume it is alpha-chain since the sequences are different, but this needs to be specified.

Yes. The RMSD is computed between the C α atoms of the structurally equivalent residues after the superposition of the two structures.

The text has been changed to clarify: “.....with an RMSD of 3.5 Å between the C α atoms of 359 structurally equivalent residues (~28%).....”

line 91: can the SPR results be shown as a supplementary figure?

Surface Plasmon Resonance data are now provided as a supplementary figure (supplementary figure 5) and a new paragraph has been added in the Methods section:

“Surface Plasmon Resonance

For binding assays experiments, a Biacore 3000 biosensor platform (GE Biosystems) equipped with a research-grade streptavidin-coated biosensor chip SA was used. The chip was docked into the instrument and preconditioned with three 1-min injections of 1 M NaCl in 50 mM NaOH. Both 3SL-PAA-biotin and 6SL-PAA-biotin (Carbosynth) oligosaccharides were injected over the second and third flow cell, respectively at 10 µg/ml diluted in HBS-P (10 mM Hepes, pH 7.4, 0.15 M NaCl and 0.005% P20). The first cell was left blank to serve as a reference. The running buffer consisted of HBS-P at a flow rate of 30 µl/min and the immobilisation levels acquired were ~160 and ~180 response units for 3SL-PAA-biotin and 6SL-PAA-biotin, respectively.

A series of diluted purified extracellular P140 and P140-P110 samples in HBS-P (1.25, 2.5, 5, 10 and 20µM) were injected over the flow cell surface at 30 µl/min. Interaction analysis were performed at 25 °C and the protein was allowed to associate and dissociate for 60 and 90 s, respectively followed by a 30 s regeneration injection step of 0.05% SDS at 30 µl/min.”

figure 2, panel f: it is not clear from the figure how the region shown participates to the heterodimer interface. Can the panel be modified so that it becomes clear how ‘f’ relates to the overview in panel ‘e’ and where the interface is.

The two panels “e” and “f” in Figure 2 have been modified. We hope that the interface between the N- and C-domains of P110 is better indicated as well as the proximity of the mutated residues to this interface.

figure 2, panels a and e: it looks from this view that the two portions that are present either in the Xray or single particle structures would clash if superimposed. Is this the case? If so this needs to be explained. If not, showing the panels from a different angle would help.

We thank the reviewer for this question, but there are no steric clashes. A new supplementary movie has been added (Supplementary movie 1) with a stereo view of the superposition of the X-ray and the Cryo-EM P140-P110 structures, where the absence of steric clashes can be cross-checked.

Reviewer #3 (Remarks to the Author):

The crystallographic parts of the project were clearly challenging and the authors should be congratulated on resolving these difficult structures.

We thank the reviewer for the kind comment.

The main conclusion of the manuscript- implied by the title- is that the Nap complex adopts 'open' and 'closed' conformational states in vivo which alternate and therefore modulate SA-mediated binding. Critically, this conclusion is based on a comparison of structures derived from single particle cryo-EM, from the in vitro reconstituted complex, and the in situ cryo-ET reconstruction (Fig 3). The detergent-solubilised Nap complex adopts a proposed 'closed' structure which is not competent for SA binding but the in situ cryo-ET density map, obtained at lower resolution, appears to show an 'open' conformation. We would presume that the latter structure is also competent for SA binding. The difficulty is that the authors then propose that both conformational states exist in vivo and alternate in some way, perhaps associated with some form of trans-membrane signalling (lines 197-200). This is highly speculative and the argument has an obvious flaw. What are the data to establish that the 'closed' conformation is not an artefact of detergent extraction or, in the case of the crystal structure, formation of the complex in vitro between the two recombinant proteins? Put another way, what is the evidence that the 'closed' state exists in vivo at all? There seems to be little direct evidence from the cryoET data.

It is indeed the case that we do not visualize the closed conformation by cryo-ET. However, this does not necessarily mean that this state is not physiological, it can also mean that it is transient or very fast. Physiologically, it would not make particular sense for a cell to carry an inactive adhesion complex for a longer period of time. In particular, when the membrane area engulfing the rod, which contains the most Naps, is so small (100x100 nm²). An additional aspect that needs to be considered here is that the bound and the ready-to bound conformations are both open. When we sub-tomogram average the Naps on the substrate side (assuming that those are mostly bound/attached) and the Naps towards the medium (assuming that those are mostly bound/attached) the structures look identical; in this regard transitioning. From the bound to unbound state needs just a short-lived mechanism of release. To visualise "in vivo", at close to atomic resolution, a molecular conformation is, of course, a major challenge that presently is seldom achieved. The finding of the "closed" conformation with different techniques (X-ray crystallography and cryo-EM from both the whole extracellular region of the P140-P110 complex and the purified Naps) together with the MAIS results and the PISA analysis (with an estimated energy of -20.5 kcal/mol and a 100% probability for the formation of the complex) provide, in our opinion, strong arguments to think that the "closed" conformation has a role in the functioning of the Nap.

We adapted the discussion to underline this point.

A second area which needs clarification are the methods used for fitting models into cryo-EM density maps. This is important for the case which the authors are making but few details are provided. It is particularly important for the fitting into the lower resolution maps shown in Fig 3. How was the fitting carried out- methods and relevant parameters (eg correlation coefficients) should be supplied? What is the evidence that the fitted models are unambiguous, unique solutions? Given the similarities between the P140 and P110 structures (lines 77-78), is it conceivable that the positions of each protein in the complex could be swapped? An argument based on a more thorough search of possible models, followed by refinement, is needed.

We thank the reviewer for this question, but there is no possibility that the position of the proteins in the complexes can be swapped. Fitting of the P110 and P140 crystal structures into the electron density map of the Nap shows a visually excellent fit, which we now show in the supplementary movies.

In addition, the quantification of fit gives correlation scores of 0.9379 and 0.94, for P110 and P140 respectively, when using a map simulated from atoms at 15Å. Due to the size difference between P110 and P140, a swapped fitting is inconceivable since the density for P140 when fitted with P110 leaves large areas of unoccupied density.

We have added in the methods that the fitting was performed using rigid-body fitting in UCSF chimera, which performs a local optimization.

A supplementary table has been added (Supplementary Table 3) containing information about the fitting quality parameters. In the main text, a brief sentence has been also included to explain the fitting of the structures into the corresponding Cryo-EM maps (line 167) :

“Rigid-body fitting of the P110-alone and P140-alone crystal structures into the electron density map of the Nap show a fit with correlation scores of 0.9379 and 0.94, respectively.”

And in the methods section, line 348:

“In all cases, rigid-body transformations were used to fit x-ray structures into the cryo-EM density maps using the fit-in-map function in UCSF Chimera.”

I would recommend that the authors reprocess their SPA without using the crystal structure as a reference (line 322). Even when filtered at 20A resolution, this can introduce bias. Using G-auto with a Gaussian blob or Warp will pick particles without bias and high degree of accuracy. Reference-free classes can be generated using a small scale hand pick and then used as a reference.

We thank the reviewer for this comment. We have performed the experiment as the reviewer has suggested by autopicking the images of the NAP complex with a Laplacian-of-Gaussian. Reprocessing showed a class with a tetramer containing 7,771 particles, and having an identical structure to the processing performed with the 3D based-autopicking. Since the

results are identical, we have not shown this data in the manuscript, but have added the sentence in the methods section:

“Cross-checking experiments with Laplacian-of-Gaussian auto-picking were performed for the NAP to verify that the structure is not reference-biased.”

Minor points

Line 78 The authors do not elaborate on the observation that the P140 and P110 proteins share similar fold topologies. A figure to illustrate this observation would be useful.

We added a supplementary figure (Supplementary figure 4) showing a side by side view of the overall structures of P110 and P140, where the similarities between both proteins are emphasized.

Line 85 Have the authors analysed the interface between P140 and P110 using the PISA protein interface server, to verify that this is the correct interface?

We have now performed a PISA interface analysis of the P140-P110N complex. The result shows an interaction surface of 2758 Å² with 30 hydrogen bonds and an estimated Gibbs free energy of -20.5 kcal/mol giving a probability value (with algorithm CSS) of 100% for the formation of the complex. A sentence has been included in the text:

“In the P140-P110 complex found by cryo-EM the interface between subunits spans 2758 Å² with 30 hydrogen bonds and an estimated Gibbs free energy of -20 kcal/mol, resulting in a 100% probability of the formation of the complex (PISA server ¹⁷).”

Lines 88-91 Where are these data? They should be included.

As suggested by the reviewer, these data have now been included by adding a supplementary figure (Supplementary figure 5).

Line 120 (and elsewhere) state the resolution criteria used.

As a standard in Relion, the 0.143 criterion is used for the resolution estimation. For the cryo-ET we have used the 0.5 criterion. We now state which criterion was used in each case in the methods section.

Lines 169-721 The binding site is not marked in Fig 3a,d, so this point is hard for the reader to visualise.

We thank the reviewer for this comment. Now in Figures 3b and e, the binding site together with the interfering loop is explicitly indicated.

Supplementary Figure 5: it is not easy to tell from the figure, but the map does not look at be at 4.1Å resolution. Could the authors include some details of the density map and secondary structure elements to substantiate this?

The resolution of this map was estimated by the FSC.

Unfortunately, the cryoEM map does not display both conformations. After analysis and classification of the dataset, only the closed conformation was present. Thus, this map was used to confirm the arrangement of the P110/P140 as solved by X-ray crystallography and to localize the positions of the C-terminal domains. Consequently, a structure with a resolution higher than that of X-ray crystallography would be needed to draw further conclusions from this structure, which given the anisotropic resolution of the density due to preferred orientation of the particles, was not feasible.

We now mention this point explicitly in the manuscript and have added in the text: “ We obtained a map with an overall resolution of 4.1 Å, although non-isotropic (Figure 2e, Supplementary Figures 7 and 8). The P140-P110N X-ray structure could be fitted as a rigid-body without modifications into the P140-P110 cryo-EM map with UCSF Chimera ¹⁶ (Supplementary Table 3 and Supplementary Figure 7). Therefore, the structure of the P140-P110 complex found by cryo-EM corresponds to the conformation of the X-ray P140-P110N structure, where access to the sialylated oligosaccharides binding site is occluded (Supplementary Movie 1 & 2).”

Reviewer #4 (Remarks to the Author):

For the two single particle structures the authors should show the following:

- Figures of a close-up density and model of the sialic acid binding site from different views to illustrate these two structures exist in a “closed” conformation.

We thank the reviewer for the constructive suggestion. For clarification: The cryo-EM map of the dimer is only used to confirm the arrangement of the P110/P140 as shown by X-ray crystallography as well to localize the position of the C-terminal domains. For both of these claims there is no need for a higher resolution (please also see the last comment to the response of Reviewer #3).

We now provide detailed movies for all structures where the fit and the transitions between the individual structures can be appreciated.

- Explanation in the main text or methods on how the model was fitted and/if refined for the 10Å map.

A supplementary table has been added (Supplementary Table 3) containing information about the fitting quality parameters. In the main text, a brief sentence has been also included to explain the fitting of the structures into the corresponding cryo-EM maps:

“The rigid-body fitting of the structures of P110 alone and P140 alone into the cryo-ET density with Chimera¹⁶ reveals major differences from the cryo-EM structure of the Nap (Figure 3, Figure 4a, Supplementary Table 3).”

And in the methods section:

“In all cases, rigid-body transformations were used to fit x-ray structures into the cryo-EM density maps using the fit-in-map function in UCSF Chimera.”

For the Cryo-ET structure at 15Å resolution:

- In figure 3e there seems to be a wider space in the density of the tight interface but the model seems to be protruding through the density and the atomic contacts are not clear. Furthermore, in figure 3d there doesn't seem to be a gap in the density at the top/extracellular interface between P140 and P110. Again, a close-up figure of this region from various views should be shown to help re-enforce the authors conclusion/suggestion of an “open” state.

We thank the reviewer for this suggestion. We now provide Supplementary Figure 12 and Supplementary movie 1 and 2 that show the positions of the P110 in the open and in the closed conformation as derived by superimposing the structures to each other.

- The clash score of the fitted atomic models for all single particle Cryo-EM and Cryo-ET experiments should be shown in the supplement to illustrate that the fitted models represent are sterically feasible.

The clash score and the correlation values of the fitted models are now provided in Supplementary Table 3.

Minor points:

- Line 133. The resolution of the Cryo-EM map of NAP in detergent is listed as 9.8Å but the FSC reports 10Å and later in the methods (line 343) it is listed as 10.5Å.

We thank the reviewer for pointing out this inconsistency. We have now made these values consistent in the text.

- Line 587 refers to supplementary figure 4 but this should be supplementary figure 5?.

We thank the reviewer for pointing this out. The text has been changed accordingly.

- Line 127: In the main text the C-terminal domain of P110 is referred to as “well defined”

but in Sup. Fig.5 it is referred to as low-resolution and flexible. Rather than “well defined” it should be referred to in the main text as clearly visible to be consistent and more accurate.

The text has been changed according to the reviewer's suggestion.

- Line 182: This density is not shown in any figures?

Density corresponding to the cytoplasmic region of the Nap was already shown in the publication of Scheffer et al. *Mol Microbiol* (2017) and there is no further improvement in this work. We have not modified the manuscript in this regard.

REVIEWERS' COMMENTS:

Reviewer #1 (Remarks to the Author):

My questions were well addressed.
Please fix small mistakes listed below before publication.

L157: (Figure 3b, e) may be (Figure 3b, c)

L232: "Subsequently, the cell extract was centrifuged at 20000 rpm at 4 °C and the supernatant applied to a 5 mL Histrap 232 column"
Specify gravity or rotor. the supernatant [was] applied.

L540: Give detailed information.

Supplementary Table 2: "Statistical significance was assessed with the paired Student's T-test."
What is another item in the pair?

Reviewer #2 (Remarks to the Author):

I am happy with the author's response. The edits and additions to the manuscript significantly improve it.

Reviewer #3 (Remarks to the Author):

The authors have responded well to my technical queries on model fitting and other more minor issues. I agree that identification of the closed state in vivo would be challenging- I was not suggesting that, rather a note of caution in using arguments based on conformations obtained from in vitro structural analysis. The revised manuscript is, overall, much improved.

Response to reviewers:

We would like to thank all reviewers for their positive comments on our manuscript. Please see our response to reviewer #1 below, as well as our point-by-point answer to the editorial requests.

Reviewer #1

L157: (Figure 3b, e) may be (Figure 3b, c)
Figure 3b is correct.

L232: Use “g” insteag of “rpm”
25000 g was added

L540: Give detailed information.

All statistical analyses were performed by comparing data of the different mutant strains with data from G37 WT strain using paired Student’s T-tests. This information has been added to the (*) the Supplementary Table 2 footnote.